# Is the Use of Intraoperative 3D Navigation for Thoracolumbar Spine Surgery a Risk Factor for Post-Operative Infection?

**DOI:** 10.3390/jcm11082108

**Published:** 2022-04-10

**Authors:** Daniel Berman, Ananth Eleswarapu, Jonathan Krystal, Henry Hoang

**Affiliations:** 1Montefiore Medical Center, Bronx, NY 10467, USA; aeleswarap@montefiore.org (A.E.); jokrysta@montefiore.org (J.K.); 2Albert Einstein College of Medicine, Bronx, NY 10461, USA; henry.hoang@einsteinmed.edu

**Keywords:** infections, complications, spine, orthopedics

## Abstract

Pedicle screw fixation is a technique used to provide rigid fixation in thoracolumbar spine surgery. Safe intraosseous placement of pedicle screws is necessary to provide optimal fixation as well as to avoid damage to adjacent anatomic structures. Despite the wide variety of techniques available, none thus far has been able to fully eliminate the risk of malpositioned screws. Intraoperative 3-dimensional navigation (I3DN) was developed to improve accuracy in the placement of pedicle screws. To our knowledge, no previous studies have investigated whether infection rates are higher with I3DN. A single-institution, retrospective study of patients age > 18 undergoing thoracolumbar fusion and instrumentation was carried out and use of I3DN was recorded. The I3DN group had a significantly greater rate of return to the operating room for culture-positive incision and drainage (17 (4.1%) vs. 1 (0.6%), *p* = 0.025). In multivariate analysis, the use of I3DM did not reach significance with an OR of 6.49 (0.84–50.02, *p* = 0.073). Post-operative infections are multifactorial and potential infection risks associated with I3DN need to be weighed against the safety benefits of improved accuracy of pedicle screw positioning.

## 1. Introduction

Pedicle screw fixation is a technique used to provide rigid fixation in thoracolumbar spine surgery [1]. There are a variety of indications for pedicle screw fixation, including correction of spinal deformity, promotion of arthrodesis, and the need to restore spinal stability if lost due to trauma, infection, or a tumor [2]. Safe intraosseous placement of pedicle screws is necessary to provide optimal fixation as well as to avoid damage to adjacent anatomic structures. A variety of techniques have been described to allow for the safe placement of pedicle screws, including the use of anatomic landmarks, fluoroscopy, ultrasound, and 3D printed models [3]. Despite the wide variety of techniques available, none thus far have been able to eliminate the risk of malpositioned screws, described in the literature as ranging from 4 to 9% [4]. Intraoperative 3-dimensional navigation (I3DN) was developed to improve accuracy in the placement of pedicle screws. Multiple prior studies have shown improved accuracy with navigated screw placement relative to freehand or fluoroscopy-assisted insertion [3,4].

Risks factors for infection during thoracolumbar fusion surgery have also been evaluated extensively in the literature. Patient-specific factors and co-morbidities, such as coronary artery disease, diabetes mellitus, male gender, ASA score, obesity, and tobacco use have all been described as risks factors for surgical site infection. Additionally, surgical factors play a large role in risk stratification. Peri-operative transfusion, operative time, number of levels fused, extension of fusion to the sacrum and pelvis, revision status, use of osteotomy, and pre-operative wound infection are all factors that have been independently associated with an increased risk of post-operative infection [5,6,7].

I3DN requires the use of intraoperative image acquisition in which the 3D fluoroscopy unit is placed circumferentially around the sterile field. This creates a theoretical risk of wound contamination. In addition, the use of I3DN also has the potential to increase operative time, which has been shown to increase infection risk in lumbar decompression and fusion surgery [8]. Lastly, to utilize I3DN, oftentimes the surgeon and other operative staff have to step away from the sterile field to shield themselves from radiation exposure, which also theoretically increases the contamination risk. To our knowledge, no previous studies have investigated whether infection rates are higher with I3DN. The purpose of this study is to compare infection rates between instrumented thoracolumbar fusions performed with I3DN versus other techniques for spinal fixation, while correcting for previously identified risk factors for post-operative infection.

## 2. Materials and Methods

A single-institution retrospective study was undertaken at our institution from January 2016 to December 2020. This start date was chosen because our institution started documenting operative data, including operative reports and surgical time, in the “Epic” electronic medical record system beginning in January 2016. The end date was chosen to allow a minimum six-month follow-up at the time of initiation of data collection. A CPT code search was used to identify all patients undergoing thoracolumbar fusion surgery by any orthopedic or neurological surgeon during this time. Based on departmental coding procedures, CPT codes 22840, 22842, 22843, 22844, 22849 were used to select our patient population. Inclusion criteria included patients over the age of 18 undergoing thoracic or lumbar fusion with instrumentation. Exclusion criteria included any fusion crossing the cervicothoracic junction, prior spine infection, age less than 18, and incomplete medical records. Cervical fusions were excluded due to a different infection risk profile for these operations as compared to thoracic and lumbar fusions as well as the lack of I3DN navigation utilization in anterior cervical fusion surgery. In addition, only adult patients were included in this analysis because of the difference in baseline infectious risk factors from the pediatric population.

Data were collected by two reviewers using the “Epic” electronic medical record system. Demographic and co-morbidity data that were collected included age, gender, body mass index (BMI), diabetes, chronic kidney disease, immunocompromised status, hypertension, alcohol/substance abuse, and smoking status (current, former, or never). In addition, surgery-specific characteristics were recorded, including use of I3DN, surgical time, use of vancomycin powder, revision status, number of levels fused, laminectomy, and interbody fusion. The primary outcome measure was the rate of return to the operating room for culture-positive incision and drainage. Regarding the primary outcome, duration from index procedure to I&D was recorded as well as the specific organism identified intra-operatively. A secondary outcome of returning to the operating room for a reason other than culture positive I&D was also recorded as well as the specific complication and the duration to revision surgery. All of these variables were identified using a combination of surgeon operative notes, operating room schedule and digitally recorded times for cases, lab reports and demographic data. Pre-operative medical assessment notes were used to identify comorbidities.

Data were then processed by the orthopedic department statistician associated with the Albert Einstein College of Medicine. All statistical analyses were performed using SAS version 9.4 (SAS Institute Inc., Cary, NC, USA). All continuous data were normally distributed. Data were assessed using Student’s t-test for continuous variables as well chi-squared and Fisher’s exact tests for categorical variables. Univariate and multivariate regression analyses were subsequently completed to identify independent risks factors for infection.

## 3. Results

After an initial CPT search, 1244 patients were initially identified for evaluation. As per the study criteria, 15 patients were excluded as they had a less than six-month follow-up, 240 for being under 18, 299 for having either a cervical fusion or a thoracic fusion crossing the cervicothoracic junction, 32 for having a pre-existing infection, and 69 for lacking an operative note identified in the medical record. After exclusion, 589 patients remained for analysis. In total, 417/589 patients underwent thoracolumbar fusion using I3DN while 172/589 patients did not utilize I3DN. All patients undergoing thoracolumbar fusion did so using posterior pedicle screw instrumentation with the use of local autograft as well as cancellous allograft chips. In total, 565/589 patients underwent fusion for degenerative conditions or deformity, while 24/589 had a fusion performed due to a metastatic disease or primary tumor of the spine. There were no significant baseline differences in the demographic characteristics between the two groups. Full demographic data are listed in Table 1.

In regard to the primary outcome measure, there was a significant difference in the rate of return to the operating room for culture-positive incision and drainage between the I3DN and non-I3DN groups (17/417 (4.9%) vs. 1/172 (0.6%), *p* = 0.025). In total, 14/18 I&Ds took place within 40 days of the index procedure while 4/18 occurred 3 months to 2 years later. In total, 17/18 I&Ds took place in patients with degenerative spinal disease or deformity and 1/18 I&Ds occurred in the I3DN group in a patient with a metastatic tumor in the spine. Specific microorganisms identified via intra-operative culture are demonstrated in Table 2. There was no significant difference in the rate of return to the operating room for reasons other than an infection between the two groups (50/417 (12%) vs. 18/172 (10.5%), *p* = 0.59).

There were significant differences in the operative data between the two groups. The I3DN cohort had a significantly longer median operative time (362.0. vs. 255.5 min, *p* < 0.0001), a higher median number of levels fused (2.0 (1.0–4.0) vs. 2.0 (1.0–3.0), *p* < 0.0001), and more prevalent use of vancomycin powder (250/417 (60%) vs. 50/172 (29.1%), *p* < 0.0001). The non-I3DN group had a significantly higher percentage of patients undergoing interbody fusion (87/172 (50.6%) vs. 162/417 (38.9%, *p* = 0.009).

In univariate analysis, three independent factors were found to be associated with the rate of return to the operating room for culture-positive I&D. These included median operative time (476.5 vs. 324, *p* = 0.0007), revision status (9/18 (50%) vs. 133/571 (23.3%), *p* = 0.02), and median levels fused (4.0 (2.0–7.0) vs. 2.0 (1.0–3.0), *p* = 0.0008). An evaluation of all demographic and operative data identified in univariate analysis is detailed in Table 3. In multivariate analysis, there were two factors found to be significantly associated with a return to the operating room for culture-positive I&D. These included revision status (OR 2.94 (1.10–7.83), *p* = 0.031) and the number of levels fused (OR 1.13 (1.02–1.27), *p* = 0.026). The relationship between the use of I3DN and rate of return to the OR for culture-positive I&D did not reach significance on multivariate analysis (OR 6.49 (0.84–50.02), *p* = 0.073). Results from the multivariate analysis are demonstrated in Table 4.

## 4. Discussion

We believe that this is the first study of its kind to evaluate I3DN as an independent risk factor for post-operative infection while statistically controlling for known risk factors. We determined that while there is a statistically significant increase in proven post-operative infection using I3DN on univariate analysis, this did not reach significance in multivariate analysis.

In the context of complex thoracolumbar fusion cases in a population with significant co-morbidities, elucidating specific risk factors for infection can be challenging. In our multivariate analysis, we found revision status and the number of levels fused to be independently associated with returning to the operating room for culture-positive I&D. In a 2012 study by Kurtz et al., they demonstrated a superficial and deep infection rate of 12.5% in revision cases versus 8.5% in primary procedures at a 10-year follow-up [9]. In addition, Janseen et al. similarly found revision status to be an independent risk for infection after instrumented spine surgery [10]. However, this was controlled for in our analysis as there was not a significant difference in revision status between the two cohorts.

On the other hand, there was a significant difference between our two study groups in terms of the number of levels fused and total surgical time. These two factors often correlate as increased fusion levels will increase the surgical time. However, a number of other factors influence surgical time as well including the complexity of anatomy, concomitant procedures such as laminectomy and interbody fusion, unforeseen complications such as durotomy, bleeding, neurologic injury, the need to revise hardware due to misplaced screws, cages, or rods, and lastly, surgeon specific efficiency. In our analysis, the reasons for the increased operative time in the I3DN group are likely multifaceted. The components of the workflow of the navigation machine, including draping the sterile field, placement of screws and sensors, image acquisition, image review, and replacement of retractors, all add to the operative time. In addition, surgeons may have been more prone to use I3DN in cases with more levels and more complex anatomy, which would add operative time regardless of the use of I3DN. A combination of these factors likely led to the statistically significant increase in operative time in the I3DN group. Similar to our analysis, Ogihara et al. specifically evaluated thoracic and lumbar fusion in a multicenter prospective study and found viea multivariate analysis that surgical time was associated with deep post-operative infection [11].

As was previously mentioned in the introduction, a number of demographic factors and co-morbidities have been shown to statistically correlate with post-operative infection in the past. In our study, they were no significant differences between the two groups in terms of any baseline demographics or co-morbidities, so these variables were unlikely to influence the outcome of this study.

The risks of the potential association between I3DN and post-operative infection must be weighed against the benefits of the procedure by the surgeons. Du et al. completed a meta-analysis of 10 articles demonstrating a significantly lower rate of pedicle violation in 3D navigation versus fluoroscopy-guided pedicle screw placement [12]. This is consistent with the findings of Mason et al., who showed in a review of 9310 pedicle screws that 3D navigation yielded an accuracy of 95% versus 68% with conventional fluoroscopy [13]. There are significant benefits to increased screw accuracy, including a possible decrease in the need for revision surgery, decreased intra-operative time revising misplaced screws, and decreased neurologic and vascular injury. These benefits were described in a retrospective review by Xiao et al., who found a statistically significant decrease in revision for hardware breakage, screw misplacement, and all-cause reoperation in thoracolumbar fusion patients treated with O-arm-assisted screw placement relative to freehand or fluoroscopy placement [14]. Avoiding revision surgery for either hardware-related complications or infection is vital to the patient, surgeon, and the health care system. Revision surgery and infection both incur significant morbidity in the patient. In addition, revisions and re-admissions are incredibly costly to both individual hospitals and, consequently, the health care system at large.

This study had several limitations. Firstly, it was a retrospective in nature based on the CPT code search through our institution’s medical records and, thus, patients were not randomized to interventions. While a prospective, randomized controlled trial would be the ideal design to reach a maximally unbiased conclusion in this study, many institution-specific limitations precluded this study design from being undertaken. In addition, surgeon-specific technical factors such as individual speed and sterility protocols are not controlled for amongst surgeons. Another important point to note is that this study was carried out at an academic institution with varying involvement of residents and fellows. The involvement of trainees in complex spine cases has the potential to influence post-operative complications, especially relating to wound closure and the possibility of superficial surgical site infection. Finally, the use of the strict criteria of culture-proven infection limited the number of events detected for our primary outcome. Nonetheless, we felt it was appropriate to use these strict criteria to prevent wound dehiscence or sterile seromas from being miscounted as infections. The total number of cases available for analysis was limited by the institutional transition to electronic medical records in 2016. For this reason, the power of the study was limited by cases performed after that date. Given the lack of statistical significance in multivariate analysis as well as baseline differences in the study groups in relation to operative time and revision status, the results of this study need to be interpreted with caution. Though the demographics of this population are relatively representative of the United States population at large, specific indications for the use of I3DN as well as surgeon-specific techniques are related to our institution and may not be representative of the average spine surgeon.

## 5. Conclusions

In conclusion, the relationship between the rate of return to the operating room for culture-positive incision and drainage and the use of I3DN is not fully elucidated given that this study did not reach significance on multivariate analysis. A follow-up study, as well as further randomized controlled trials with adequate power, needs to be completed to fully determine this relationship. In the complex and multifactorial subject of risk factors for post-operative infection, the risks and benefits of I3DN need to be critically evaluated pre-operatively by the surgeon.

## Figures and Tables

**Table 1 jcm-11-02108-t001:** Demographic and clinical characteristics by I3DN use in 589 patients.

		I3DN Use N = 417	No I3DN Use N = 172	*p* Value
Age mean ± SD	–	57.8 ± 12.9	57.7 ± 13.3	0.96
Gender n(%)	Female	255(61.2)	115(66.9)	0.19
	Male	162(38.8)	57(33.1)	
Return to OR for I&D n(%)	No	400(95.9)	171(99.4)	0.025
	Yes	17(4.1)	1(0.6)	
Return to OR for other reason n(%)	No	366(88.0)	154(89.5)	0.59
	Yes	50(12.0)	18(10.5)	
Surgical time median(IQR)	–	362.0(279.0–466.0)	255.5(210.0–349.0)	<0.0001
BMI median(IQR)	–	29.8(25.3–33.9)	31.0(26.1–35.0)	0.1
Use of vancomycin power n(%)	No	167(40.0)	122(70.9)	<0.0001
	Yes	250(60.0)	50(29.1)	
Revision status n(%)	No	322(77.2)	125(72.7)	0.24
	Yes	95(22.8)	47(27.3)	
Levels fused median(IQR)	–	2.0(1.0–4.0)	2.0(1.0–3.0)	<0.0001
Laminectomy n(%)	No	42(10.1)	19(11.1)	0.72
	Yes	375(89.9)	153(88.9)	
Interbody fusion n(%)	No	254(61.1)	85(49.4)	0.009
	Yes	162(38.9)	87(50.6)	
Diabetes n(%)	No	287(69.0)	123(71.5)	0.54
	Yes	129(31.0)	49(28.5)	
Chronic kidney disease n(%)	No	387(92.8)	164(95.3)	0.25
	Yes	30(7.2)	8(4.7)	
Immunocompromise n(%)	No	385(92.6)	165(95.9)	0.13
	Yes	31(7.4)	7(4.1)	
Hypertension n(%)	No	158(38.0)	65(37.8)	0.97
	Yes	258(62.0)	107(62.2)	
Alcohol/substance abuse n(%)	No	380(91.4)	149(86.6)	0.08
	Yes	36(8.6)	23(13.4)	
Smoking n(%)	Never	226(54.5)	94(54.6)	0.55
	Former	120(28.9)	55(32.0)	
	Current	69(16.6)	23(13.4)	

**Table 2 jcm-11-02108-t002:** Specific microorganisms identified based on intra-operative culture.

O-Arm Use	Microorganism
Yes	Klebsiella pneumoniae, finegoldia magna
Yes	MRSA
Yes	Proteus Mirabilis, Prevotella Bivia
No	Proteus Mirabilis
Yes	MSSA, Corynebacterium striatum
Yes	MRSA
Yes	Enterococcus Faecalis, Streptococcus Mitis, Prevotella Bivia
Yes	Enterobacter Cloacae
Yes	MSSA, Proteus Mirabilis
Yes	VRE (Vanco resistant enterococcus)
Yes	E Faecalis, E Coli
Yes	Pseudomonas
Yes	Staph Lugdenesis, Proteus Penneri, Enterococcus
Yes	Klebsiella Pneumoniae
Yes	Enterobacter Cloacae
Yes	MSSA
Yes	MSSA
Yes	Klebsiella Pneumoniae, Proteus Mirabalis

**Table 3 jcm-11-02108-t003:** Univariable associations of demographics and clinical variables associated with return to OR for I&D.

		Return to OR for I&D (N = 18)	Did not Return to OR for I&D (N = 571)	*p* Value
Age mean ± SD	–	62.0 ± 10.5	57.6 ± 13.1	0.16
Gender n(%)	Female	12(66.7)	358(62.7)	0.73
	Male	6(33.3)	213(37.3)	
Return to OR for other reason n(%)	No	13(72.2)	507(89.0)	0.046
	Yes	5(27.8)	63(11.0)	
Surgical time median(IQR)	–	476.5(363.0–655.0)	324.0(250.0–426.0)	0.0007
BMI median(IQR)	–	28.2(22.0–31.7)	30.0(25.9–34.0)	0.12
Use of vancomycin power n(%)	No	9(50.0)	280(49.0)	0.94
	Yes	9(50.0)	291(51.0)	
Revision status n(%)	No	9(50.0)	438(76.7))	0.02
	Yes	9(50.0)	133(23.3)	
Levels fused median(IQR)	–	4.0(2.0–7.0)	2.0(1.0–3.0)	0.0008
Laminectomy n(%)	No	4(22.2)	57(10.0)	0.11
	Yes	14(77.8)	514(90.0)	
Interbody fusion n(%)	No	11(64.7)	328(57.4)	0.55
	Yes	6(35.3)	243(42.6)	
Diabetes n(%)	No	11(61.1)	399(70.0)	0.42
	Yes	7(38.9)	171(30.0)	
Chronic kidney disease n(%)	No	18(100)	533(93.4)	0.62
	Yes	0(0)	38(6.6)	
Immunocompromise n(%)	No	18(100)	532(93.3)	0.62
	Yes	0(0)	38(6.7)	
Hypertension n(%)	No	10(55.6)	213(37.4)	0.12
	Yes	8(44.4)	357(62.6)	
Alcohol/substance abuse n(%)	No	18(100)	511(89.6)	0.24
	Yes	0(0)	59(10.4)	
Smoking n(%)	Never	12(66.7)	308(54.1)	0.42
	Former	5(27.8)	170(29.9)	
	Current	1(5.5)	91(16.0)	

**Table 4 jcm-11-02108-t004:** Factors associated with return to OR for I&D on multivariable logistic regression.

Characteristic		Adjusted Odds Ratio (95% CI)	*p* Value
I3DN use	Yes	6.49 (0.84–50.02)	0.073
	No (reference)	1	
Revision status	Yes	2.94 (1.10–7.83)	0.031
	No (reference)	1	
Levels fused	–	1.13 (1.02–1.27)	0.026

## Data Availability

Not applicable.

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
