# Peer review of "Is the Use of Intraoperative 3D Navigation for Thoracolumbar Spine Surgery a Risk Factor for Post-Operative Infection?"

_jcm, 2022, doi:10.3390/jcm11082108_

Round 1

Reviewer 1 Report

Title and abstract

We donot agree with the title, as for reasons mentioned below, i don't agree that the observations of this study indicate that the very use of 3d navigation was associated with infections... It may be reworded as a risk factor for infections Abstract is reasonably well written
Intro The intro is well focused and confusing the study purpose lucidly   Methods and results Retrospective model is not a good methodology for the purpose of this study. As the authors have shown, the patients who underwent 3d navigation had longer levels of fusion and underwent interbody fusion..  In other words, these patients had relatively more complex pathologies and surgeries. This bias of this study cannot be ignored. Thus the need for or use of navigation can be considered as a risk factor for injection. However, the use of 3d navigation itself may not be directly associated with infection (based on this model)   The results are not clear. Multivariate analysis does not show any significant relationship between i and d and 3d navigation. There is no multivariate analysis for infection/ positive culture and 3d navigation. The conclusion that 3d navigation increases infection is not substantiated   Discussion The underlying diagnosis and actual surgeries performed need to be added, as these are relevant factors influencing infection The discussion is very superficial. It seems to graze upon topics related to infections and navigation. I would recommend that the authors stick to the complications or challenges related to navigation in spine surgery and then elaborate the findings of the current study   The limitations of the study need to be elaborated   For reasons previously mentioned, i don't agree with the conclusion of the study and therefore recommend that it may be modified    

Author Response

  • The title has been modified to “risk factor” for infection as opposed to “associated”. The reviewer’s points are well-taken considering the lack of significance at the .05 level on multivariate analysis
  • The authors agree that in an ideal world, a prospective, randomized-controlled trial would be the best methodology for this study and other orthopedic spine surgery studies. However, given the power needed to establish significance as well as the extreme difficulty with randomization and prospective patient follow-up, as well as the ethical concerns associated with this and individual surgeon preference for navigation versus no navigation, it was not possible for this specific study.  Unfortunately, a retrospective analysis was the best study design that was feasible at our institution.
  • The conclusions and results have been reworded to state that the conclusion about the association between I3DN and infection did not reach significance on multivariate analysis
  • In regards to surgeries performed, a line was clarified to state that all thoracolumbar fusions were performed using posterior pedicle screw instrumentation with the use of local autograft and allograft. The authors believe that the specifics of the remainder of the surgeries are adequately addressed in table 3.  This describes and compared use of laminectomy, interbody fusion, revision status, and use vancomycin powder between the two groups.
  • In regards to diagnosis, the number of patients with degenerative spinal disease, deformity, and metastatic tumors to the spine were clarified.
  • The authors appreciate the reviewer concerns in regards to the discussion. However, we presented the discussion in this manner for what we believe is a very important reason.  Infection us a complex and multi-faceted outcome, and we felt it necessary to address the risk factors in relation to spine surgery specifically, and then to tie this back to data from our study.  We believe that due to the lack of baseline differences in terms of risk factors in our patient population, this ultimately controlled for our outcome very well.  We also thought it was necessary to describe the benefits of navigation and elaborate on that literature, as ultimately any conclusion regarding infection risk will contribute to a cost benefit analysis by the operative surgeon given the documented benefits of navigation in regards to accuracy of instrumentation.
  • In regards to the limitations section, a description was added about both study design and reasoning for the number of cases as our institution being limited by certain factors outside of the authors’ control

Reviewer 2 Report

Reviewer Comments

Thank you very much for the opportunity to review the manuscript submission entitled: Is the Use of Intraoperative 3D Navigation for Thoracolumbar Spine Surgery Associated with an Increased Risk of Infection?

The current paper aims to compare infection rates between instrumented thoracolumbar fusions performed with I3DN versus other techniques for spinal fixation, while correcting for previously identified risk factors for post-operative infection. The data is interesting, and it has a relevant rationale, however, some limitations and constructive comments are pointed below:

Specific comments

Title and Abstract

  • Include study design in the title
  • Include patient characteristics in the abstract (n, mean age)
  • Provide in the abstract an informative and balanced summary of what was done and what was found.
  • Include MeSH terms as keywords

Introduction

  • The scientific background and rationale for the investigation need to be emphasized.
  • The hypothesis of the study needs to be stated.

Methods

  • All the methods section should be discussed in detail.
  • The inclusion/exclusion criteria should be more detailed and described.
  • Clearly define all outcomes.
  • Describe any efforts to address potential sources of bias.
  • Can you explain how the study's sample size was determined?

Statistical methods and results

  • What is the basis for the selection of statistical tests? Did the data (study variables) follow normal distribution?
  • Stick to the table specifications format, which is outlined in the author's instructions.

Discussion

  • Give a cautious overall interpretation of results considering objectives, limitations, the multiplicity of analyses, and results from similar studies.
  • Discuss the generalizability (external validity) of the study results.

Author Response

  • The study design is described multiple times in the abstract and the body of the manuscript. The authors did not believe it necessary to include in the title as this is not required by the journal guidelines.  However, if all reviewers believe this would improve the manuscript, we are happy to add this to the end of the title.  Reviewers 1 and 3 did not mention this as a potential issue with the manuscript
  • The authors appreciate the suggestions related to the abstract, though the journal specific word-limit precludes inclusion of more extensive discussion of patient characteristics or study. We believe that the major points are currently highlight in the abstract, including summary, study design, primary outcome, and conclusion.  Further elaboration is limited by word limit.
  • Thank you for the suggestions in relation to the MeSH keywords. These have now been modified in the manuscript.
  • We believe the scientific background of risk factors for post-operative spine infection are extensively elaborated upon in the introduction. In addition, rationale and benefits for the use of navigation are discussed as well.  In the last paragraph of the introduction, the purpose of the study is stated clearly.  The authors did not provide a hypothesis “for” or “against” infection as to not seem biased and let the data ultimately speak for itself.
  • Thank you for the comment regarding the inclusion and exclusion criteria. Two sentences were added to clarify rationale for this.
  • We believe that potential bias was accounted for by describing the multivariate regression analysis technique
  • The primary and secondary outcome measures are well defined in this section
  • The beginning of the first paragraph of the methods section addresses how the sample size for this study was selected
  • The continuous data was normally distributed. A sentence was added to clarify this is the last section of methods.  The statistical tests used are the academic standard for normally distributed continuous data and categorical data.
  • A multivariate regression analysis was used to control for other patient-related and surgical variables for post-operative infection
  • Upon review, the tables seem to comply with the specifications at set out by the author instructions and the other two reviewers did not see any issues. Please elaborate on what specifically needs to be changed to improve the table formatting.
  • This is the first study of its kind specifically in relation to I3DN and post-operative infection, so comparison to similar studies is limited. We believe the remainder of the discussion emphasizes previous studies in relation to risk factors for post-operative infection in spine surgery as well as the previously documented benefits of I3DN.
  • A section was added at the end of limitation in the discussion to address the generalizability as well as cautious interpretation of results.

Reviewer 3 Report

I would like to congratulate the authors on their well-done work. However, given the study design, it is difficult to discern whether the cause of the increase in infections is the I3DN, or whether it is due to the surgical time. This makes it necessary to consider a future study with a more ambitious design to be able to discriminate the time variable in the origin of the infections. 

I believe that the conclusion should be less ambitious since the origin of the increase in the number of infections is not fully clarified by this type of study. 

Author Response

  • Thank you for these comments and suggestions. This is in line with the other reviewers comments about the conclusions and they have been modified to more accurately represent the statistical conclusions of the study.

Round 2

Reviewer 1 Report

Most of the concerns raised previously have been accepted. 

I congratulate the authors for the study.

Reviewer 2 Report

The authors have satisfactorily addressed all the queries raised by me.